# Facile and general electrochemical deuteration of unactivated alkyl halides

Pengfei Li[1], Chengcheng Guo[1], Siyi Wang[1], Dengke Ma[1], Tian Feng[1], Yanwei Wang[1] & Youai Qiu [1✉]

Herein, a facile and general electroreductive deuteration of unactivated alkyl halides (X = Cl, Br, I) or pseudo-halides (X = OMs) using $D_2O$ as the economical deuterium source was reported. In addition to primary and secondary alkyl halides, sterically hindered tertiary chlorides also work very well, affording the target deuterodehalogenated products with excellent efficiency and deuterium incorporation. More than 60 examples are provided, including late-stage dehalogenative deuteration of natural products, pharmaceuticals, and their derivatives, all with excellent deuterium incorporation (up to 99% D), demonstrating the potential utility of the developed method in organic synthesis. Furthermore, the method does not require external catalysts and tolerates high current, showing possible use in industrial applications.

[1] State Key Laboratory and Institute of Elemento-Organic Chemistry, Frontiers Science Center for New Organic Matter, College of Chemistry, Nankai University, 94 Weijin Road, Tianjin 300071, China. ✉email: qiuyouai@nankai.edu.cn

Compounds with D-labeling play an important role in the realm of chemistry, mechanistic studies, and pharmaceutical science[1–9]. Dehalogenative deuteration from organic halides is a straightforward way to obtain deuterated target compounds. However, major studies focused on dehalogenative deuteration of aryl halides and activated alkyl halides[10–20]. In contrast, corresponding reports on unactivated alkyl halides, particularly on unactivated alkyl chlorides, have not been well elucidated. Several realizations have been reported but suffer from a number of problems, such as limiting to alkyl iodides using radical initiators[21,22], the usage of a stoichiometric metal reductant[23], long reaction time, and low reactivity of tertiary alkyl halides using photocatalysis[24], etc. Therefore, the development of simple, efficient, and environmentally-friendly approaches to access deuterated compounds from unactivated alkyl halides with high D-incorporation is highly desirable.

Meanwhile, organic electrochemistry has become a powerful tool for sustainable synthesis by employing electron instead of stoichiometric amounts of redox reagents in the past decade[25–44]. Organic halides are important players in traditional organic chemistry as well as promising electrochemistry, owing to their versatile reactivity and readily available properties. Thus, impressive electrochemical progress of organic halides has been made, such as reductive bifunctionalization of alkenes[45–48], reductive coupling with halides[49–56], and others[57–68]. Hence, electrochemical deuteration of organic halides could be a straightforward and promising route for valuable deuterated compounds, which has gained increasing attention[69–71]. However, current research and success in electrochemical deuteration have been focused on aryl halides. In sharp contrast, electrochemically driven dehalogenative deuteration of unactivated halides has proven elusive (Fig. 1a). Direct electrochemical deuteration of unactivated alkyl halides is challenging due to their extremely negative reduction potential[19,72–74], and likelihood of competitive undesired side reactions[50].

Herein, we report our effort in developing this dehalogenative deuteration of unactivated alkyl halides, making it a general, economically and environmentally approach through electrochemistry (Fig. 1b). Notable features of this strategy include: (a) broad scope and excellent D-incorporation, including unsatisfactory tertiary chlorides in previous studies, (b) $D_2O$ as an economical deuterium source, (c) external catalysts-free and the use of electricity as an environmentally-friendly reductant, (d) catalytic amounts of electrolyte and no sacrificial anode, (e) tolerance with high current (up to 500 mA), making it potentially applicable in industrial production, (f) late-stage deuteration of complex natural products and drug derivatives.

## Result

**Optimization of electroreductive dehalogenation deuteration.** Initially, we started our investigation using $D_2O$ as the economical deuterium source, unactivated alkyl bromide **1** as the substrate, and DMF as the solvent. After careful optimization, the corresponding product **2** was finally obtained in excellent yield and D-incorporation under 30 mA constant current with carbon felt (CF) and Pb as electrodes, in the presence of DIPEA as well as the catalytic amount of TBAI as the electrolyte in an undivided cell at room temperature for 10 hours (Table 1, entry 1). The control

**(a) State-of-the-art in electrochemical deuteration of organic halides**

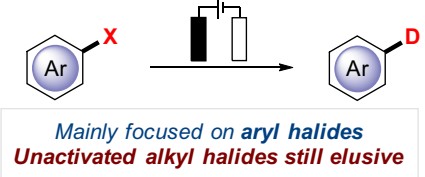

Mainly focused on **aryl halides**
*Unactivated alkyl halides still elusive*

**(b) This work: electrochemical deuteration of unactivated alkyl halides**

Alkyl—X $\xrightarrow[\text{D}_2\text{O}]{\text{CF} \quad \text{Pb}}$ Alkyl—D

**X = Cl, Br, I, OMs**
**Alkyl = 1°, 2°, 3°**

*65 examples
up to 99% D-incorporation*

- General scope of alkyl halides, including 3° alkyl chlorides, in high D-level
- $D_2O$ as economical deuterium source
- No sacrificial anode
- Direct electroreduction-external catalyst free
- Compatible with high current
- Late stage deuteration of complex natural products and drug derivatives

from Ibuprofen
**55**, 72%, 98% D

from D-mannofuranose
**63**, 79%, 99% D

*500 mA high current finished in 15 min*

**Fig. 1 Electrochemical deuteration of organic halides. a** State-of-the-art in electrochemical deuteration of organic halides. **b** A facile and general electrochemical deuteration of unactivated alkyl halides.

experiment revealed that the electricity was essential for the observed reactivity (entry 2). The reaction still worked well in the absence of DIPEA with 60% yield (entry 3), which showed that DMF could also be oxidized on anode. In addition, reducing the current to 20 mA led to lower yield and D-incorporation (entry 4). Surprisingly, the desired transformation proceeded smoothly with a higher current at 100 mA in 60 min (entry 5). It is notable that even under an extremely high current at 500 mA, the reaction still proceeded well to give the target product in only 15 min, without diminishing the yield and D-labeling level, showing potential usage in industrial amplification (entry 6). DMSO was proven to be a poor solvent for this transformation (entry 7), while MeCN gave a comparable yield but lower D-incorporation (entry 8). The reaction also worked well when $^{n}Bu_4NBF_4$ was used as the electrolyte, albeit in a slightly lower yield (entry 9). Finally, different electrodes were tested but none of the results surpassed entry 1 (entries 10-13).

**Substrate scope.** With the optimized reaction conditions in hand, we probed the generality of this electroreductive deuteration reaction. Firstly, various alkyl bromides were screened (Fig. 2A). Alkyl bromides with different lengths of carbon chains worked well to afford the corresponding products with excellent D-incorporation (2-7). Substituents with electron-withdrawing or donating properties were compatible and showed little effect on the reaction (8-13). Besides, other aromatic rings, including naphthalene, indole, and dihydrobenzofuran (15-17), were all well tolerated. The success of this procedure could be confirmed by the excellent compatibility of a wide range of functionalities, such as terminal ether (18), ester (19), and thioether (20), unprotected lactam (21-23), Boc-protected amine (24), terminal olefin (25), tetrahydropyran (26), as well as internal ester (27). Interestingly, for substrates containing both Br

**Table 1 Screening of reaction conditions[a].**

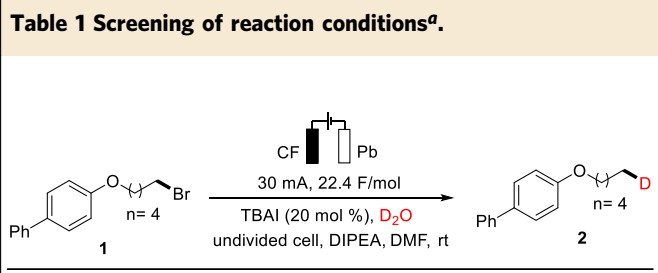

| Entry | Variation | Yield %[b] | D - inc %[c] |
|---|---|---|---|
| **1** | **None** | **96** | **99** |
| 2 | No electricity | N.D. | 0 |
| 3 | No DIPEA | 60 | 99 |
| 4 | 20 mA | 85 | 94 |
| 5 | 100 mA (60 min) | 94 | 99 |
| 6 | 500 mA (15 min) | 90 | 99 |
| 7 | DMSO | 38 | 63 |
| 8 | MeCN | 99 | 89 |
| 9 | $^{n}Bu_4NBF_4$ | 85 | 99 |
| 10 | CF (+) \| CF (−) | 23 | 87 |
| 11 | CF (+) \| Ni (−) | trace | 0 |
| 12 | CF (+) \| Pt (−) | 7 | 90 |
| 13 | Pt (+) \| Pt (−) | 25 | 75 |

[a]Reaction conditions: alkyl halide **1** (0.5 mmol), $D_2O$ (25.0 mmol), TBAI (20 mol%), DIPEA (1.5 mmol), DMF (5.0 mL) under 30 mA constant current in an undivided cell at room temperature for 10 hours with carbon felt as anode and lead plate as cathode.
[b]Isolated yield of product.
[c]Deuterium incorporation determined by [1]H NMR. CF carbon felt, DIPEA N, N-Diisopropylethylamine, TBAI $^{n}Bu_4NI$, N.D. not detected.

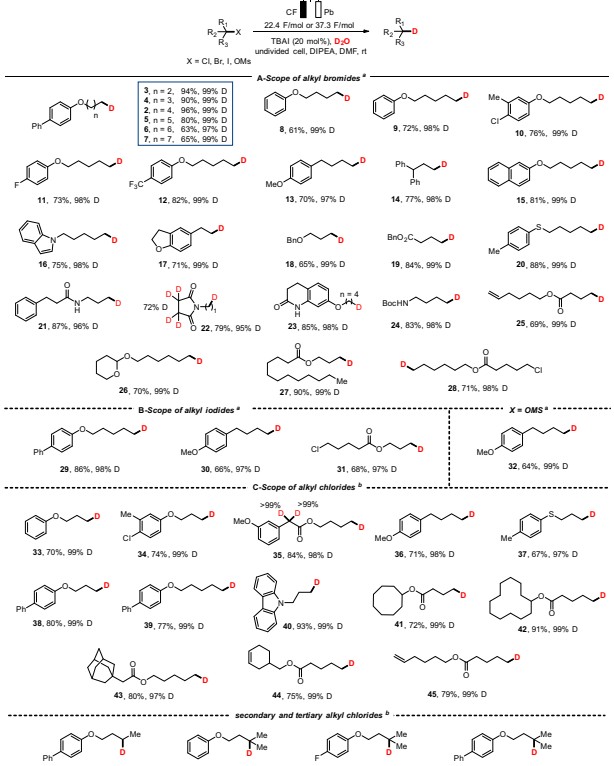

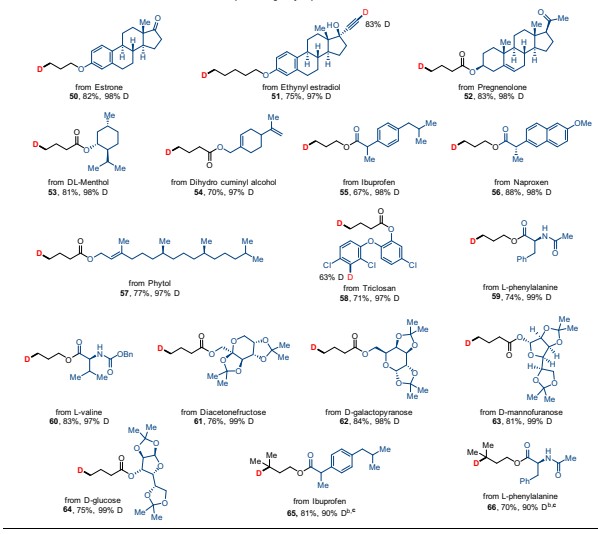

**Fig. 2 Electrochemically driven dehalogenative deuteration of unactivated alkyl halides.** Electrochemically driven dehalogenative deuteration of unactivated alkyl halides. **A** Scope of alkyl bromides, **B** scope of alkyl iodides, **C** scope of alkylchlorides, and **D** complex biologically important substrates. [a]Reaction conditions: bromide, iodide or pseudo-halides (0.5 mmol), $D_2O$ (25 mmol), TBAI (20 mol %), DIPEA (1.5 mmol), DMF (5.0 mL) under 30 mA constant current in an undivided cell at room temperature for 10 h with carbon felt as anode and lead plate as cathode. Isolated yield of product. Deuterium incorporation determined by [1]H NMR. [b]Alkyl chlorides (0.5 mmol) was conducted under 50 mA constant current with graphite felt as anode. [c]Deuterium incorporation determined by GC-MS.

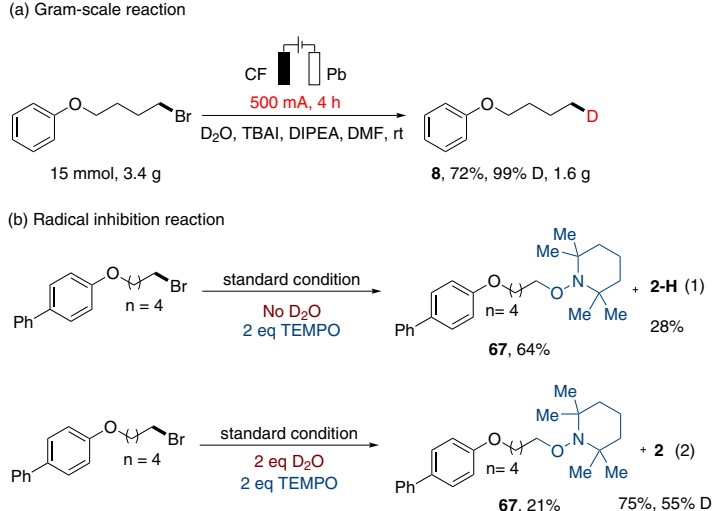

**Fig. 3 Fast electrochemically driven dehalogenative deuteration under high current.** [a]Reaction conditions: alkyl bromide (0.5 mmol), D$_2$O (25.0 mmol), TBAI (20 mol %), DIPEA (1.5 mmol), DMF (5.0 mL) under 500 mA constant current in an undivided cell at room temperature for 15 minutes with carbon felt as anode and lead plate as cathode. Isolated yield of product. [b]Alkyl chlorides (0.5 mmol) was conducted with graphite felt as anode.

**Fig. 4 Gram-scale reaction and mechanistic studies. a** The compound **8** could be prepared in gram-scale with higher isolated yield. Gram-scale reaction. **b** Radical inhibition reaction was performed, indicating that the alkyl radical might be involved.

and Cl, it was found that the C-Br bond was easier to cleave (**28**). Besides, alkyl iodides also worked smoothly in this protocol, affording the corresponding products with excellent deuterium incorporation (Fig. 2B, **29-31**). We were pleased to find that OMs could be a suitable leaving group to give the target product in 64% yield and 99% D-incorporation (**32**).

Next, we turned our attention to a more challenging target, unactivated alkyl chlorides (Fig. 2C). To our delight, primary alkyl chlorides took part in the reaction with a broad range of functionalities and showed similar results to alkyl bromides, yielding the corresponding products in good efficiency and D-incorporation (**33-45**). Furthermore, other alkyl chlorides, especially tertiary substrates that did not work well in previous studies[24], yielded the desired products unquestionably in satisfactory D-incorporation using this method (**46-49**). To further demonstrate the generality, and environmentally-friendly nature of the developed method, as well as its application

in bioactive molecules and drug discovery, the late-stage deuteration was conducted using a series of natural products, pharmaceuticals, and their derivatives (Fig. 2D). The dehalogenative deuteration of pharmaceuticals including Estrone (**50**), Pregnenolone (**52**), DL-Menthol (**53**), Phytol (**57**), and Triclosan (**58**) was successfully achieved in 77–83% yield with 97-98% D-incorporation. Pharmaceutical intermediates ethynyl estradiol (**51**) and dihydro cuminyl alcohol (**54**) containing unsaturated bonds can also be used in this protocol. Ibuprofen (**55**) and naproxen (**56**), which are commonly used for the ease of pain, afforded the corresponding deuterated compounds in good yields (67%, 88%) with excellent deuterium incorporation (both 98% D). Furthermore, amino acid and glucose derivatives that widely exist in organisms can also be deuterated smoothly with high levels of D-incorporation (**59−64**, 97−99%). In addition, the tolerance of the reaction system to tertiary chlorides can be extended to complex substrates (**65**, **66**).

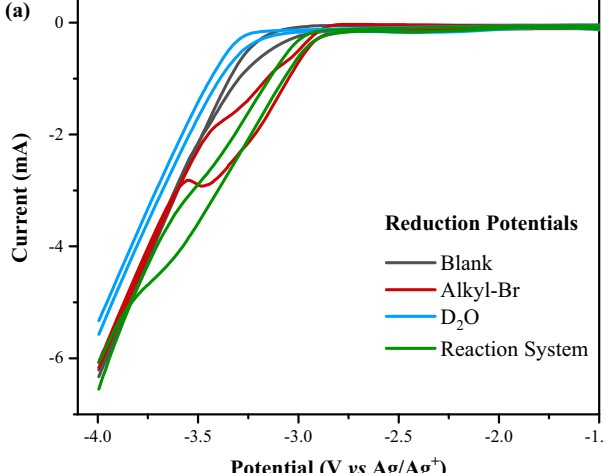

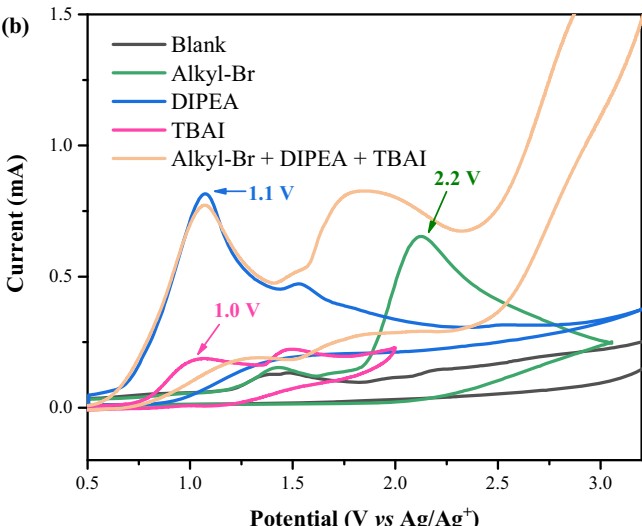

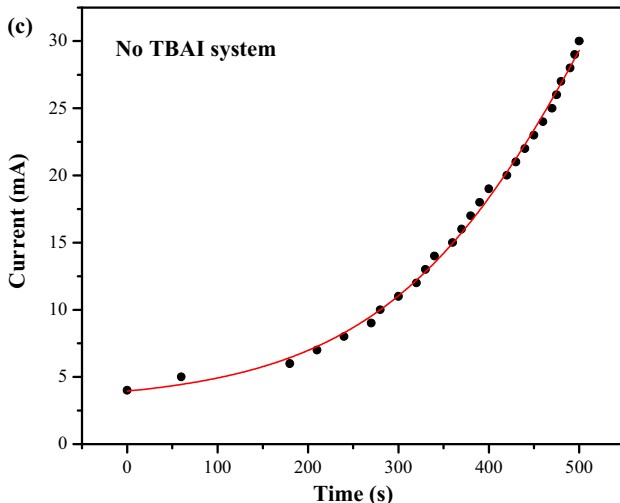

**Fig. 5 Cyclic voltammetry experiments.** Using glass carbon as working electrode, Pt plate and Ag/Ag⁺ as counter and reference electrode. (Related compounds (5 mM) in MeCN containing 0.1 M ⁿBu₄NBF₄, 100 mVs⁻¹). **a** Reduction potentials. **b** Oxidant potentials. **c** Variation curve of current without TBAI.

**Electrochemical dehalogenative deuteration of alkyl bromides with 500 mA current**. As high current density has a key impact on the amplification of reaction scales, we further explored the practicability of some representative substrates at a current of 500 mA. To our delight, the reactions were complete during a much shorter reaction time (in only 15 minutes). Simple alkyl bromides with a variety of functional groups (**2**, **16**, **23**, and **26**) as well as complex alkyl bromides derived from ibuprofen (**55**), L-phenylalanine (**59**), Diacetonefructose (**61**), and D-mannofuranose (**63**) were all amenable. level of D-incorporation has not changed at all (Fig. 3). Alkyl chloride was also well tolerated and the corresponding product **37** was obtained in 15 min. In addition, this electrochemical reduction protocol could be applied to the gram-scale preparation of deuterated product **8** under 500 mA constant current (Fig. 4a).

**Mechanistic studies**. Several mechanistic studies were conducted to gain insight into the reaction mechanism. Firstly, 2 equiv. of TEMPO was added to the reaction system and in the absence of D₂O, the TEMPO adduct was obtained in 64% yield (**67**), together with hydrodehalogenated product **2-H** in 28% yield (Fig. 4b, eq 1). Then, repeat the reaction with 2 equiv. of D₂O resulted in the TEMPO adduct in 21% yield and deuterodehalogenated product **2** in 75% yield with 55% D-incorporation (Fig. 4b, eq 2). These results indicated that the alkyl radical might be involved in the electrochemical system.

Then, we sought to gain further insight into the reaction mechanism through cyclic voltammetry (CV) experiments. A reduction peak of alkyl bromides **1** at −3.40 V (vs Ag/Ag⁺ in CH₃CN) was observed (Fig. 5a, the red line), and no obvious reduction peak of D₂O was observed (Fig. 5a). At the same time, the oxidation potential of TBAI (Fig. 5b, pink line, 1.0 V) was very close to that of DIPEA (Fig. 5b, blue line, 1.1 V). Thus we inferred that both TBAI and DIPEA might act as the electron donors in this reaction. Notably, the anodic oxidation of alkyl bromide **1** was also indicated with higher potential than that of DIPEA (green line). This might explain the improved yield once external DIPEA was added by inhibiting the undesired oxidation or decomposition of the starting material (Table 1, entries 1 and 3). In addition, during the reaction system without TBAI, an increase in current can be clearly observed, indicating the electrolysis of DIPEA incorporated with formed bromo anion may further generate conductive substances (Fig. 5c).

Based on the mechanistic studies and previous literature[5,48], a plausible mechanism is proposed (Fig. 6). The reaction was initiated by the anodic oxidation of DIPEA and/or TBAI or DMF and the direct reduction of alkyl halides on the cathode to form the corresponding alkyl radical **I** and halide anions. The radical **I** continued to be reduced to an alkyl anion **II** at the cathode. Finally, intermediate **II** reacted with deuterated water to provide the expected product **III**. At the same time, iminium ion intermediate **IV** generated from DIPEA on the anode might combine with the released halogen anions to form ammonium salt, a possible conductive species.

## Discussion

In conclusion, we have developed a general dehalogenative deuteration of unactivated alkyl (pseudo)halides (X = Cl, Br, I, OMs) driven by electrochemical force with high reactivity, selectivity, and excellent D-incorporation. D₂O was used as an efficient and economical deuterium source. A wide range of unactivated alkyl halides containing diverse functionalities are well tolerated, as well as the extension to late stage deuteration of natural products, pharmaceuticals, and their derivatives. We envisage that the native features, including tolerance of high current and

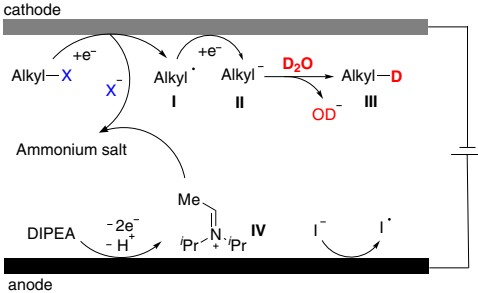

**Fig. 6 A plausible mechanism.** Cathodic reactions converting organic halides to deuterated product. Anodic oxidation of DIPEA and/or TBAI or DMF was occurred on anode.

environmentally-friendly conditions, will bring further opportunity in its application to organic synthesis, drug discovery and modification, and even industrial production. Further electrochemical transformations of unactivated alkyl halides are currently ongoing in our laboratory.

## Methods
**General procedure of the electrochemical dehalogenative deuteration of alkyl bromide, iodide or pseudo-halides.** The electrocatalysis was carried out in an undivided cell with a carbon felt anode (10 mm × 15 mm × 5 mm) and a lead cathode (10 mm × 15 mm × 0.3 mm). To a 15 mL pre-dried undivided electrochemical cell (15 mL) equipped with a magnetic bar were added alkyl bromide, iodide or pseudo-halides (0.5 mmol, 1 equiv), TBAI (36.9 mg, 0.1 mmol, 20 mol%), and DMF (5.0 mL). Then $D_2O$ (25 mmol, 50 equiv) and DIPEA (1.5 mmol, 3.0 equiv) were added via a syringe. The electrocatalysis was performed at room temperature with a constant current of 30 mA maintained for 10 h. The carbon felt anode was washed with EtOAc (3 × 5 mL) in an ultrasonic bath. $H_2O$ (20 mL) was added to the system, and the resulting mixture was extracted with EtOAc (3 ×20 mL). The combined organic phase was dried with anhydrous $Na_2SO_4$, filtered, and concentrated in vacuo. The crude product was purified by column chromatography to furnish the desired product.

**General procedure of the electrochemical dehalogenative deuteration of alkyl chlorides.** The electrocatalysis was carried out in an undivided cell with a graphite felt anode (10 mm × 15 mm × 5 mm) and a lead cathode (10 mm × 15 mm × 0.3 mm). To a 15 mL pre-dried undivided electrochemical cell (15 mL) equipped with a magnetic bar were added alkyl chloride (0.5 mmol, 1 equiv), TBAI (36.9 mg, 0.1 mmol, 20 mol%) and DMF (5.0 mL). Then $D_2O$ (25 mmol, 50 equiv) and DIPEA (1.5 mmol, 3.0 equiv) were added via a syringe. The electrocatalysis was performed at room temperature with a constant current of 50 mA maintained for 10 h. The carbon felt anode was washed with EtOAc (3 × 5 mL) in an ultrasonic bath. $H_2O$ (20 mL) was added to the system, and the resulting mixture was extracted with EtOAc (3 × 20 mL). The combined organic phase was dried with anhydrous $Na_2SO_4$, filtered, and concentrated in vacuo. The crude product was purified by column chromatography to furnish the desired product.

**General procedure of the electrochemical dehalogenative deuteration of alkyl bromides and chloride with 500 mA current.** The electrocatalysis was carried out in an undivided cell with a carbon felt anode (or graphite felt) (10 mm × 15 mm × 5 mm) and a lead cathode (10 mm × 15 mm × 0.3 mm). To a 15 mL pre-dried undivided electrochemical cell (15 mL) equipped with a magnetic bar were added alkyl bromide or chloride (0.5 mmol, 1 equiv), TBAI (36.9 mg, 0.1 mmol, 20 mol%) and DMF (5.0 mL). Then $D_2O$ (25 mmol, 50 equiv) and DIPEA (1.5 mmol, 3.0 equiv) were added via a syringe. The electrocatalysis was performed at room temperature with a constant current of 500 mA maintained for 15 min (reaction system exothermic). The carbon felt anode was washed with EtOAc (3 × 5 mL) in an ultrasonic bath. $H_2O$ (20 mL) was added to the system, and the resulting mixture was extracted with EtOAc (3 × 20 mL). The combined organic phase was dried with anhydrous $Na_2SO_4$, filtered, and concentrated in vacuo. The crude product was purified by column chromatography to furnish the desired product.

**Gram-scale synthesis of 8.** To an undivided reaction flask (diameter: 40 mm, length: 130 mm, volume: 200 mL) equipped with a teflon-coated magnetic stirring bar and teflon cap, a carbon felt anode (25 mm × 50 mm × 5 mm), and a lead cathode (25 mm × 50 mm × 0.3 mm) were added (4-bromobutoxy)benzene (3.44 g, 15 mmol), TBAI (1.11 g, 3.0 mmol), DMF (100 mL), $D_2O$ (15 mL, 0.75 mol) and DIPEA (5.80 g, 45 mmol). The electrocatalysis was performed at room temperature with a constant current of 500 mA maintained for 4 h (the reaction system was exothermic). The carbon felt anode was washed with EtOAc (3 × 20 mL) in an

ultrasonic bath. $H_2O$ (200 mL) was added to the system, and the resulting mixture was extracted with EtOAc (3 × 200 mL). The combined organic phase was dried with anhydrous $Na_2SO_4$, filtered, and concentrated in vacuo. The crude product was purified by column chromatography to furnish the desired product.

## Data availability
The authors declare that the data supporting the findings of this study are available within the article and its Supplementary Information files. Extra data are available from the author upon request.

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

## Acknowledgements

Financial support from the Fundamental Research Funds for the Central Universities (No. 63213063), Frontiers Science Center for New Organic Matter, Nankai University (Grant No. 63181206) and Nankai University are gratefully acknowledged.

## Author contributions

Y.Q. and P.L. conceived and designed the study and wrote the manuscript. P.L, C.C.G, S.W, D.M, T.F, Y.W, Y.Q. performed the experiments, mechanistic studies and revised the manuscript. All authors contributed to the analysis and interpretation of the data.

## Competing interests

The authors declare no competing interest.
