## [Peer Review File · Nature Communications]

Editorial Note: Parts of this peer review file have been redacted as indicated to avoid copyright infringement.

REVIEWER COMMENTS

Reviewer #1 (Remarks to the Author):

Compounds with D-labeling usually represent essential functions in the realm of reaction mechanism and pharmaceutical chemistry. Moreover, efficient protocol is still urgently required for the construction of D-labeled molecules. In this manuscript, Qiu and co-workers reported a novel strategy for the construction of D-labeled alkanes by electrochemical dehalogenative deuteration of unactivated alkyl halides with D₂O. The substrate scope is fairly broad and many of sensitive functional groups are tolerated under the deep-reductive conditions. Besides, catalytic amounts of electrolyte and non-sacrificial anode are used in this protocol, which is a progress for the current electroreductive manners. Moreover, high current is also fit to the reaction and has little effect on the amplification, which reveals a big potential for the industrial application. For these reasons, the paper deserves to be published in Nat. Commun. after the following revisions.

1. In Scheme 4b, where is the H atom from in the product 2-H?
2. In page 4, Line 1. The definite article “a extremely” should be revised to “an extremely”.
3. Page 1, line 4 of Introduction, the spelling “acitivated” should be revised.
4. The amount of deuterium water is greatly excessive (50 equivalence). We suppose that much of D₂O is wasted and electrolyzed on the cathode or anode. How about the results when using lower amounts of D₂O? The screening results of deuterium water should be provided.
5. In Table 1, the cathodic material seems to be essential to the transformation. Did the authors try other kinds of cathodic material? Why does Pb cathode meet to the optimal result? Please try to explain.
6. In Table 1, have the authors tried other kinds of reducing agents except DIPEA? Screening results should be provided.
7. For the substrate scope, many of sensitive functional groups like -F, -Cl, -CF₃, -NHBoc and alkene are well tolerated under the deep-reductive conditions. In general, such functional groups are not compatible under the deep-reductive electrolyzed conditions, can the authors explain this phenomenon? Moreover, the aryl chlorides should show better reactivity, but the aryl carbon-chloride bond was preserved in product 31, which should be explained by the authors.

8. Have the authors tried out the other pseudo halides, including tosylates, triflates or mesylates?
9. The results of compound 32 & 55 are different from other products. How is the reaction going for the two substrates?
10. In the footnote b of Scheme 2, graphite felt anode is used instead of carbon felt anode. What's difference between the two kinds of anode and how does this affect the reaction? To my knowledge, there should not be big difference.
11. How about other electrophilic reagents? A few examples should be tested to increase the impact of this work or to be reported in another work.
12. The consumed charge of most substrates is rather huge (22.4 F/mol for alkyl bromides/iodides and 37.3 F/mol for alkyl chlorides). What made the electron waste in this transformation? However, the consumed charge is reduced to 9.3 F/mol when high current (500 mA) is preformed to the reaction. What's going on when the electric current increases? Consumed charge should be provided in SI along with the reaction time.
13. In Scheme 5, DIPEA acts as the sacrificial reducing agent and is oxidized on the anode. However, we speculate that it can not exclude the anodic oxidation of TBAI since I⁻ is easier oxidized than DIPEA. Evidence of I⁻ oxidation could be verified and the proposed mechanism might be revised.
14. Some reports and reviews involving electrochemical deuteration should be cited, including Chem. Commun., 2022, DOI: 10.1039/D2CC00344A; J. Am. Chem. Soc. 2022, 144, 2062; Sci. Bull. 2021, 66, 2412.

Reviewer #2 (Remarks to the Author):

This manuscript presents the electrochemical dehalogenative deuteration of alkyl halides which occurs in mild reaction conditions and is applicable to a wide substrate scope. If the deuteration of aryl halides has been well established since the pioneering work of Murray, Renaud and Grimshaw in the 70's, the related electrochemical deuteration of alkyl halides remains elusive.

Despite interesting results, several important points concerning this electrochemical method remain to be clarified.

The exact role of the supporting electrolyte is vaguely discussed. Indeed, iodides are typically oxidized at a redox potential lower than that of DIPEA. Yet it seems that it has been overlooked by

the authors. Intriguingly, all cyclic voltammetry (CV) studies were performed in the absence of an iodide source, which casts doubts upon the validity of the reaction mechanism which was proposed. If it is true that optimization studies showed that TBABF₄ could also be employed as electrolyte, all the scope was investigated with TBAI. Hence it is of utmost importance to clarify the role of this iodide source. In the same vein, considering that the bromide ion is also prone to get oxidized at redox potentials close to that of DIPEA, it is regrettable that its impact was not investigated by CV. Moreover, chlorides are oxidized at a slightly higher potential -about 400 mV higher than bromides- but considering the current which is applied during this electrolysis (500 mA), it seems difficult to rule out their concurrent oxidation. Of note, it would have been of high interest to monitor the kinetic of the reaction process in the presence as well as in the absence of halogen sources, which may have clarified the role of the released halides.

The addition of DIPEA (1 equiv) increased the reaction yield by 30%. This is significant but it means that the reaction can also proceed with an alternative oxidation process at the anode as without amine the product was obtained in 60% yield. Once again, the mechanistic proposal undermines this key experimental observation. Besides, the optimization of the amount of added DIPEA should be more detailed as it is not clear why the authors chose to employ 1 equiv.

Importantly, given that the reaction is performed in an undivided cell and that the oxidation of DIPEA can lead to an alpha-amino radical, a halogen atom transfer (XAT) process must be considered. The group of Leonori (*Science* 2020, 367, 1021) recently reported that such radical species (generated by photoredox catalysis) are prone to perform XAT on alkyl halides and that this strategy can be applied to the very same dehalogenative deuteration process. The reactivity of these alpha-amino radicals has been known for long as described by Neumann (*Synthesis* 1987, 665) or by Landais and coll (*Chem Rev* 2018, 6516). It is regrettable that there is no mention of this seminal work despite its clear significance.

Oddly, the authors report a galvanostatic (constant current) process and they claim that the current intensity increases when the reaction is performed without TBAI. Can the authors comment thoroughly on this point as such scenario is not compatible with a galvanostatic setup but with a potentiostatic one? The constant current applied is really high (500 mA) and it is surprising that the authors did not observe the consequent formation of by-products since, with such a current value, the supporting electrolyte and the solvent should not remain unchanged (even the ammonium salt could be reduced) except if the surfaces of the electrodes are important. This data is of high importance and must be clearly discussed to avoid any reproducibility issues.

Of note, the paragraph concerning the use of high constant current to reduce the reaction time that could be of industrial interest is written twice in the manuscript.

To conclude, considering the lack of electrochemical rationale combined with insufficient mechanistic evidences, I think that this manuscript does not meet the requirement for a publication in *Nature Communication*. In addition to properly cite literature, this work should be thoroughly reviewed by an electrochemist. Additionally, before any resubmission of this work, I would advise the authors to rewrite the manuscript by a native English speaker.

Reviewer #3 (Remarks to the Author):

In this manuscript, Qiu and coworkers reported an expedient and general electrochemical deuteration of unactivated alkyl halides using D₂O as the economic deuterium source by cathodic reduction manner. This work shows a broad substrate scope with excellent efficiency and deuterium incorporation, including primary and secondary alkyl halides, as well as steric tertiary chlorides, which normally are relative hash reaction site. The electroreductive D-incorporation method developed by the authors is significant and useful in the academic and industry of medicinal discovery, to be an alternative protocol, avoiding the usage of stoichiometric metal reductant and harsh conditions. To prove this, the authors highlighted the late-stage dehalogenative deuteration of natural products, pharmaceuticals, and their derivatives, with ideal deuterium incorporation, and conditions tolerated with external catalysts free and high current. Several mechanistic studies were performed and the possible pathway was proposed, however, more detail and evidence should be discussed and needed. Thus, based on the significance and novelty of the whole work, I recommend its publication in Nature Communications after addressing following comments.

1. In the mechanistic studies, the role of TBAI should be needed to further clarify. The authors claimed that the TBAI mainly used as electrolyte, however, does the authors measure the CV of TBAI, it is also possible to be oxidized at the anode to generate iodine (I₂), then react with alkyl chloride or alkyl bromide to facilitate the reaction.
2. In the substrate scope, it seems all the substrates need the hetero-atom, especially the ester linker, so how would be the amide linker going, which would more interesting and significant in drug discovery.
3. The products 32 and 48 showed more than one D-incorporation reaction site, compound 32 is in benzylic site, and 48 is in the terminal of alkyne, which are very sensitive and reactive. It is easy made readers confused, so the authors should better clarify the starting materials with halogen's position or number in the manuscript or SI.
4. In the schemes, the reaction conditions of undivided cell were recommended to include as well as the amount of electricity or efficiency of electricity.
5. In the manuscript, please revise some spelling of the words with quotes, such as "usage of 'stiochiometric' metal reductant; in high D-incorporation is still highly 'desirale'." "Initially, we start our investigation using D₂O as the economic deuterium source," should be better revised to past tense.

Comments from the reviewers:

Reviewer 1

Question 1: In Scheme 4b, where is the H atom from in the product 2-H?

Response: Thanks for the comments. According to reported literatures, sacrificial reagents (*J. Am. Chem. Soc.* **1981**, *103*, 1172.; *Chem. Eur. J.* **2019**, *25*, 6911.) and solvents (*Chem. Sci.* **2020**, *11*, 10414.) in the reaction system might act as hydrogen donors. In our reaction, it might be DIPEA or DMF. To conclude it, we conducted the reaction using D-DMF as the solvent under argon atmosphere and finally no D-incorporation observed in the obtained product. Thus we speculated that the H atom of the 2-H product was provided by DIPEA.

Question 2: In page 4, Line 1. The definite article “a extremely” should be revised to “an extremely”.

Response: Thanks for the comments. We have revised “a extremely” to “an extremely” for the whole manuscript.

Question 3: Page 1, line 4 of Introduction, the spelling “acitivated” should be revised.

Response: Thanks for the comments. The spelling of "activated" has been corrected throughout the article.

Question 4: The amount of deuterium water is greatly excessive (50 equivalence). We suppose that much of D₂O is wasted and electrolyzed on the cathode or anode. How about the results when using lower amounts of D₂O? The screening results of deuterium water should be provided.

Response: Thanks for the comments. Reaction using less D₂O gave the target product in comparable yields while in lower deuterium incorporations, eg, 30 equivalent in slightly lower 95% D-incorporation and 10 equivalent in 84% D-incorporation (see entries 2-3 in the table below). Considering the importance and potential application of compounds in satisfactory D-incorporation, we finally decided the loading of D₂O in current conditions. The relevant results have been given in the Supporting Information as entries 2-3 in Table S1.

Entry	Variation	Yield % ^b	D - inc % ^c
1	None	96	99
2	30 eq D ₂ O	91	95
3	10 eq D ₂ O	94	84

^a Reaction conditions: undivided cell, carbon felt anode, lead plate cathode, constant current = 30 mA, alkyl halide **1** (0.5 mmol), D₂O (50.0 equiv), TBAI (20 mol%), DIPEA (3.0 equiv), DMF (5.0 mL), rt, 10 h. ^b Yield of isolated product. ^c Deuterium incorporation determined by ¹H NMR. DIPEA = *N,N*-Diisopropylethylamine, TBAI = ⁿBu₄NI.

Question 5: In Table 1, the cathodic material seems to be essential to the transformation. Did the authors try other kinds of cathodic material? Why does Pb cathode meet to the optimal result? Please try to explain.

Response: Thanks for the comments. We additionally tried Fe, Cu, Nb as cathodic materials besides GF, Ni, and Pt as cathode materials that have been provided in the original submission. As a result, almost no product could be detected when Fe and Nb were used as cathodes. While Cu cathode afforded the product in slightly lower D-incorporation (see the table below). These results have been put in Table S1 as entries 4-6 in the SI.

Entry	Variation	Yield % ^b	D - inc % ^c
1	None	96	99
2	Fe (-)	trace	0
3	Cu (-)	97	97
4	Nb (-)	trace	0

^a Reaction conditions: undivided cell, carbon felt anode, lead plate cathode, constant current = 30 mA, alkyl halide **1** (0.5 mmol), D₂O (50.0 equiv), TBAI (20 mol%), DIPEA (3.0 equiv), DMF (5.0 mL), rt, 10 h. ^b Yield of isolated product. ^c Deuterium incorporation determined by ¹H NMR. DIPEA = *N,N*-Diisopropylethylamine, TBAI

= ⁿBu₄NI.

We speculated that the higher hydrogen evolution potential of Pb might be the reason for this result.

[redacted]

Question 6: In Table 1, have the authors tried other kinds of reducing agents except DIPEA? Screening results should be provided.

Response: Thanks for the comments. Experiments of DBU, Et₃N, PPh₃, NPh₃ instead of DIPEA working as reducing agents, as well as Zn and Mg working as sacrificial anodes have been performed. As a result, DBU, Et₃N, and NPh₃ could give almost the same D-incorporation while with a little bit lower yields (entries 2-4 in the table below), except PPh₃, giving much lower yield as 52% (entry 5 in the table below). The relevant results have been given in the Supporting Information as entries 7-12 in Table S1. Meanwhile Zn and Mg anodes didn't afford improved results (entries 6-7 below).

Entry	Variation	Yield % ^b	D - inc % ^c
1	None	96	99
2	DBU	89	99
3	Et ₃ N	94	99
4	NPh ₃	94	99
5	PPh ₃	52	95
6	Zn (+)	38	63
7	Mg (+)	99	89

^a Reaction conditions: undivided cell, carbon felt anode, lead plate cathode, constant current = 30 mA, alkyl halide **1** (0.5 mmol), D₂O (50.0 equiv), TBAI (20 mol%), DIPEA (3.0 equiv), DMF (5.0 mL), rt, 10 h. ^b Yield of isolated product. ^c Deuterium

incorporation determined by ^1H NMR. DIPEA = *N,N*-Diisopropylethylamine, TBAI = $^n\text{Bu}_4\text{NI}$.

Question 7: For the substrate scope, many of sensitive functional groups like -F, -Cl, -CF₃, -NHBoc and alkene are well tolerated under the deep-reductive conditions. In general, such functional groups are not compatible under the deep-reductive electrolyzed conditions, can the authors explain this phenomenon? Moreover, the aryl chlorides should show better reactivity, but the aryl carbon-chloride bond was preserved in product **31**, which should be explained by the authors.

Response: Thanks for the comments. The functional groups like -F, -Cl, -CF₃, -NHBoc and alkene were well tolerated under current conditions, perhaps because of their high reduction potential comparing to the cleaved C-X bond in the substrate, some of which were also observed in the electrochemical arylation of alkyl halides (*J. Am. Chem. Soc.* **2021**, *143*, 12985.).

For substrate **31**, the alkoxy groups on the *para*-position of aromatic ring as well as the electron donating methyl group on the *ortho*-position increased the reduction potential of the aryl C-Cl bond, which make it untouched. Competition experiments using **31a** and **31** (1:1) as substrates giving no C(sp²)-Cl bond cleaved product supported the hypothesis.

Question 8: Have the authors tried out the other pseudo halides, including tosylates, triflates or mesylates?

Response: Thanks for the advice. We tried the following substrates containing pseudo-halides, such as OMs, OTs, OTf, OAc, and OPiv. We were pleased to find that OMs could be a suitable leaving group to give the target product **32** in 64% yield and 99% D-incorporation under current conditions. We are still working hard to expand the scope of leaving group to illustrate the generality. In addition, we have added the results in the manuscript and SI.

Question 9: The results of compound 32 & 55 are different from other products. How is the reaction going for the two substrates?

Response: Thanks for the comments. The new deuterium sites appearing in compounds 32 and 55 were due to the electron-withdrawing groups on both sides of this position, which enhanced the Lewis acidity and made the C-H bond active.

Question 10: In the footnote b of Scheme 2, graphite felt anode is used instead of carbon felt anode. What's difference between the two kinds of anode and how does this affect the reaction? To my knowledge, there should not be big difference.

Response: Thank you for the comments. The elements composition of graphite felt and carbon felt is both carbon. While graphite felt could be formed from carbon felt at a higher temperature, which might be one of the reason to the fact that the graphite felt has higher purity and lower ash content. Thus we speculated that graphite felt should have better conductivity and reactivity (*Int J Energy Res.* 2020, 44, 3839).

Question 11: How about other electrophilic reagents? A few examples should be tested to increase the impact of this work or to be reported in another work.

Response: Thanks for the comments. We have tried several electrophiles as followed. At this moment, preliminary results with benzaldehyde and benzyl acrylate as electrophiles were obtained, and the target products were afforded in 35% and 23% yield, respectively. Further optimization of current results and exploration suitable conditions for different kinds of electrophiles are still underway in our group, we hope relevant work would be reported in our future work.

Question 12: The consumed charge of most substrates is rather huge (22.4 F/mol for alkyl bromides/iodides and 37.3 F/mol for alkyl chlorides). What made the electron

waste in this transformation? However, the consumed charge is reduced to 9.3 F/mol when high current (500 mA) is performed to the reaction. What's going on when the electric current increases? Consumed charge should be provided in SI along with the reaction time.

Response: Thanks for the comments. We examined the charge consumption at different time periods by measuring the conversion of the reaction (Figure 1 below). The charge consumption was only 4.5 F/mol in the 0-2 hour period (Figure 2 below). As the reaction went by, the decrease in halide concentration affected the reaction efficiency and might lead to electron waste in this transformation. In fact, the reaction could complete within eight hours. The reaction time was extended to 10 h to make sure all kinds of substrates could reach full conversion. Gladly, the stability of the formed products were not touched. High current density could accelerate the reaction process when high current (500 mA) was utilized, yielding reduced charge consumption. In addition, we have added the consumed charge information in the SI.

Figure 1. Reaction monitoring. (Determined by ^1H NMR using CH_2Br_2 as an internal standard)

Figure 2. Efficiency of electricity.

Question 13: In Scheme 5, DIPEA acts as the sacrificial reducing agent and is oxidized on the anode. However, we speculate that it can not exclude the anodic oxidation of TBAI since I⁻ is easier oxidized than DIPEA. Evidence of I⁻ oxidation could be verified and the proposed mechanism might be revised.

Response: Thanks for the professional suggestions. We conducted the CV study of TBAI. As measured, the oxidation potential of TBAI (pink line, 1.0 V) is very close to that of DIPEA (blue line, 1.1 V). Control experiment in the manuscript (entry 3, Table 1) also showed that in the absence of DIPEA, the reaction also gave the product in 60% yield, indicated that the TBAI could act as electron donor. Thus we inferred that both TBAI and DIPEA could be oxidized and act as electron donor simultaneously in the reaction system. We also revised the mechanism part by adding TBAI as electron donor.

Cyclic voltammetry studies

Question 14: Some reports and reviews involving electrochemical deuteration should be cited, including Chem. Commun., 2022, DOI: 10.1039/D2CC00344A; J. Am. Chem. Soc. 2022, 144, 2062; Sci. Bull. 2021, 66, 2412.

Response: Thanks for the comments. The above literatures have been cited as ref. [7], [8], and [40], respectively. We also carefully checked the related references, and other recent literatures were also added as references:

9. Kopf, S., Bourriquen, F., Li, W., Neumann, H., Junge, K., Beller, M. Recent Developments for the Deuterium and Tritium Labeling of Organic Molecules. *Chem. Rev.* **122**, 6634–6718 (2022).

41. Zhang, B., Gao, Y., Hioki, Y., Oderinde, M. S., Qiao, J. X., Rodriguez, K. X., Zhang, H.-J., Kawamata, Y., & Baran, P. S. Ni-Electrocatalytic C(sp³)-C(sp³) Doubly Decarboxylative Coupling. *Nature*, DOI: doi.org/10.1038/s41586-022-04691-4. (2022).

53. Zhang, W., Lu, L., Zhang, W., Wang, Y., Ware, S. D., Mondragon, J., Rein, J., Strotman, N., Lehnher, D., See, K. A., & Lin, S. Electrochemically driven cross-

electrophile coupling of alkyl halides. *Nature* **604**, 292–297 (2022).

Reviewer 2

Question 1: The exact role of the supporting electrolyte is vaguely discussed. Indeed, iodides are typically oxidized at a redox potential lower than that of DIPEA. Yet it seems that it has been overlooked by the authors. Intriguingly, all cyclic voltammetry (CV) studies were performed in the absence of an iodide source, which casts doubts upon the validity of the reaction mechanism which was proposed. If it is true that optimization studies showed that TBABF₄ could also be employed as electrolyte, all the scope was investigated with TABI. Hence it is of utmost importance to clarify the role of this iodide source. In the same vein, considering that the bromide ion is also prone to get oxidized at redox potentials close to that of DIPEA, it is regrettable that its impact was not investigated by CV. Moreover, chlorides are oxidized at a slightly higher potential -about 400 mV higher than bromides- but considering the current which is applied during this electrolysis (500 mA), it seems difficult to rule out their concurrent oxidation. Of note, it would have been of high interest to monitor the kinetic of the reaction process in the presence as well as in the absence of halogen sources, which may have clarified the role of the released halides.

Response: Thanks a lot for the professional comments.

About the role of iodide source from the electrolyte: We did the CV studies of TABI and measured its oxidation potential. As we can see, the oxidation potential of TABI (pink line, 1.0 V) was very close to that of DIPEA (blue line, 1.1 V). Thus we inferred that both TABI and DIPEA might act as the electron donors in this reaction. In fact, control experiment in the manuscript (entry 3, Table 1) indicated that in the absence of DIPEA, the target product could still be obtained in 60% yield, supporting the hypothesis of TABI working as electron donor. We have modified the description accordingly in the mechanism part of the manuscript.

Cyclic voltammetry studies

On the other hand, it came the possibility that the in situ generated I_2 from the anodic oxidation of TBAI might react with alkyl halides to form corresponding alkyl iodide, making the cleavage of C-X bonds easier. To address this concern, we did the following reactions. Firstly, replacing TBAI to TBABF₄ made no difference to the results, indicated that TBAI was not an essential factor in this reaction (eq 1-2 in the scheme below). Secondly, mixing the alkyl bromide substrate with I_2 in DMF at rt for 12 h gave no alkyl iodide product as detected by HRMS (eq 3 below). Meanwhile, the alkyl iodide could be obtained only around 38% yield in the reaction of alkyl bromide, TBAI, and DIPEA in DMF stirred at rt for 12 h (eq 4 below), while still no transformation for alkyl chloride under the same conditions (eq 5 below). Considering the high efficiency of alkyl bromides and chlorides in our reaction, we would say the in situ generation of alkyl iodide should be less likely.

About the role of the released halides from the substrate: We conducted kinetic studies using substrate (4-bromobutoxy)benzene with varied loading of NaBr (0.2 equivalent, 0.5 equivalent and 1.0 equivalent) as an external halogen source under standard conditions. From current results in hand, it seemed that the additionally added halogen source made no positive or even negative effect on the reaction rate. When 1 equiv. of NaBr was added, the final yield even decreased (see figure below). Yeah, taking the released halides into consideration for its function as electron donor and even

kinetic effects is indeed a novel idea, we really appreciate it. While the reaction system is quite complex, we assumed that the released halides might either combine with TBA^+ from TBAI or combine with iminium ion from anodic oxidation of DIPEA as the counter anion to form new electrolyte. That's why we can use only catalytic electrolyte in the reaction.

Concerning released chloride anion under high current (500 mA), we newly conducted this experiment although it was not provided in our first submission. Finally, the alkyl chloride was also well tolerated and the corresponding product **37** was obtained in 15 min. Since the reaction under high current (500 mA) was very fast (finished in 15 min), it was difficult to conduct the kinetic studies with exact data. Anyway, we have added the results of alkyl chloride under 500 mA current in the manuscript.

Question 2: The addition of DIPEA (1 equiv) increased the reaction yield by 30%. This is significant but it means that the reaction can also proceed with an alternative oxidation process at the anode as without amine the product was obtained in 60% yield.

Once again, the mechanistic proposal undermines this key experimental observation. Besides, the optimization of the amount of added DIPEA should be more detailed as it is not clear why the authors chose to employ 1 equiv.

Response: Thanks for the kind suggestion. As answered above, we have improved the mechanism part by adding anodic oxidation of TBAI as one possible electron donor pathway. Besides, different equivalents of DIPEA were screened. Finally, as the decreasing amount of DIPEA, the yield dropped. When 1 equiv. of DIPEA was used, the yield was just 66% (see table below).

Entry	Variation	Yield % ^b	D - inc % ^c
1	None	96	99
2	1 eq DIPEA	66	99
3	2 eq DIPEA	93	99

^a Reaction conditions: undivided cell, carbon felt anode, lead plate cathode, constant current = 30 mA, alkyl halide **1** (0.5 mmol), D₂O (50.0 equiv), TBAI (20 mol%), DIPEA (3.0 equiv), DMF (5.0 mL), rt, 10 h. ^b Yield of isolated product. ^c Deuterium incorporation determined by ¹H NMR. DIPEA = *N,N*-Diisopropylethylamine, TBAI = ⁿBu₄NI.

Question 3: Importantly, given that the reaction is performed in an undivided cell and that the oxidation of DIPEA can lead to an alpha-amino radical, a halogen atom transfer (XAT) process must be considered. The group of Leonori (Science 2020, 367, 1021) recently reported that such radical species (generated by photoredox catalysis) are prone to perform XAT on alkyl halides and that this strategy can be applied to the very same dehalogenative deuteration process. The reactivity of these alpha-amino radicals has been known for long as described by Neumann (Synthesis 1987, 665) or by Landais and coll (Chem Rev 2018, 6516). It is regrettable that there is no mention of this seminal work despite its clear significance.

Response: It's a very good suggestion. Indeed, we also have this concerns during the investigations. Since the observation and trapping of in situ generated alpha-amino radical is not so easy, we chose to use amines without alpha hydrogen, such as NPh₃, that could not form alpha amino radical, to rule out the possibility. As we can see, when NPh₃ was used, the reaction proceeded smoothly without change of the yield and D-incorporation. Thus the XAT pathway should be less likely.

Entry	Variation	Yield % ^b	D - inc % ^c
1	None	96	99
4	NPh ₃	94	99

^a Reaction conditions: undivided cell, carbon felt anode, lead plate cathode, constant current = 30 mA, alkyl halide **1** (0.5 mmol), D₂O (50.0 equiv), TBAI (20 mol%), DIPEA (3.0 equiv), DMF (5.0 mL), rt, 10 h. ^b Yield of isolated product. ^c Deuterium incorporation determined by ¹H NMR. DIPEA = *N,N*-Diisopropylethylamine, TBAI = ⁿBu₄NI.

Question 4: Oddly, the authors report a galvanostatic (constant current) process and they claim that the current intensity increases when the reaction is performed without TBAI. Can the authors comment thoroughly on this point as such scenario is not compatible with a galvanostatic setup but with a potentiostatic one?

Response: Electrocatalysis was performed in constant current mode using a HSPY-36-03 potentiostat with a maximum display voltage of 37 V (Figure 3). The galvanostatic mode could not directly reach the target current of 30 mA without adding electrolyte in the reaction system (*J. Am. Chem. Soc.* 2022, 144, 2343). Due to the limitation of voltage, the initial current of the reaction was only 4 mA (Figure 4). With the increase amount of ammonium salt produced in the reaction, the current reached to 30 mA after 10 minutes, which could be regarded as a constant voltage reaction process in the first 10 minutes. After that, the reaction mode was a galvanostatic process, and the voltage decreased slowly with the continuously increased amount of the ammonium salt. In addition, we have added the results in the SI.

Figure 3 Reaction system

Figure 4 Without TBAI

Question 5: The constant current applied is really high (500 mA) and it is surprising that the authors did not observe the consequent formation of by-products since, with such a current value, the supporting electrolyte and the solvent should not remain unchanged (even the ammonium salt could be reduced) except if the surfaces of the electrodes are important. This data is of high importance and must be clearly discussed to avoid any reproducibility issues.

Response: Our reaction showed good mass balance and functional group tolerance. When stopped at 10 min for the 500 mA high current experiment, 71% of the product was obtained and 27% of the raw material was recovered, indicating the stability of the substrates in the reaction system. High current was also applied with good working efficiency in one recent publication (*J. Am. Chem. Soc.* **2021**, *143*, 12985.).

The concept of current refers to the rate of electron movement, which facilitates redox processes (*Acc. Chem. Res.* **2020**, *53*, 72–83). Voltage might be a key factor affecting the stability of compounds, and the voltage (13.9 V) under high current conditions (500 mA) is not scaled up so much according to the ratio of current (8.7 V at 30 mA) (Figures 4 and 5), which might be the reason that we did not detect by-products under 500 mA conditions.

Figure 5 500 mA Condition

Question 6: Of note, the paragraph concerning the use of high constant current to reduce the reaction time that could be of industrial interest is written twice in the manuscript.

Response: The paragraph concerning “the use of high constant current to reduce the reaction time that could be of industrial interest” have been modified.

Question 7: Additionally, before any resubmission of this work, I would advise the authors to rewrite the manuscript by a native English speaker.

Response: The manuscript has been checked and revised as suggested by several of our former colleagues, who are also working in chemistry and being native English speakers. We hope the revised manuscript could be better.

Reviewer 3

Question 1: In the mechanistic studies, the role of TBAI should be needed to further clarify. The authors claimed that the TBAI mainly used as electrolyte, however, does the authors measure the CV of TBAI, it is also possible to be oxidized at the anode to generate iodine (I₂), then react with alkyl chloride or alkyl bromide to facilitate the reaction.

Response: Thanks for the comments. CV studies showed that the oxidation potential of TBAI (1.0 V) was very close to that of DIPEA (1.1 V). Taking the control experiment without DIPEA (60% yield, entry 3, Table 1) in the manuscript into consideration, it could be concluded that both TBAI and DIPEA should be oxidized and act as electron donors in this reaction.

Concerning the reaction of alkyl halides with in situ generated I₂, we did the following reactions. Firstly, using TBABF₄ instead of TBAI brought no difference, showing non-essential role of TBAI in the reaction (eq 1-2 below). Then mixing alkyl bromide and I₂ in DMF and stirring at rt for 12 h afforded no alkyl iodide product (eq 3 below). Meanwhile, the alkyl iodide could be obtained in around 38% yield by the reaction of alkyl bromide, TBAI, and DIPEA in DMF stirred at rt for 12 h (eq 4 below), while still no transformation for alkyl chloride under the same conditions (eq 5 below). Considering the high reactivities and yields of alkyl bromides and chlorides in our reaction, the generation of alkyl iodide in situ should be less likely.

Cyclic voltammetry studies

Question 2: In the substrate scope, it seems all the substrates need the hetero-atom, especially the ester linker, so how would be the amide linker going, which would more interesting and significant in drug discovery.

Response: Thanks for the comments. Several amide substrates were tested under the standard conditions, and the results showed that amide substrates could work smoothly to afford the corresponding products with good efficiency and high D-incorporation. The newly added examples (**21**, **22**) have been added in the manuscript and with fully characterized data in the Supporting Information.

Question 3: The products 32 and 48 showed more than one D-incorporation reaction site, compound 32 is in benzylic site, and 48 is in the terminal of alkyne, which are very sensitive and reactive. It is easy made readers confused, so the authors should better clarify the starting materials with halogen's position or number in the manuscript or SI.

Response: Thanks for the comments. We have included the starting material with multiple deuterated sites in the Supporting Information.

Question 4: In the schemes, the reaction conditions of undivided cell were recommended to include as well as the amount of electricity or efficiency of electricity.

Response: Thanks for the comments. The reaction conditions of undivided cell and efficiency of electricity have been modified in the whole manuscript.

Question 5: In the manuscript, please revise some spelling of the words with quotes, such as “usage of ‘stiochiometric’ metal reductant; in high D-incorporation is still highly ‘desirale’.” “Initially, we start our investigation using D₂O as the economic deuterium source,” should be better revised to past tense.

Response: Thanks for the comments. We have corrected the wrong sentence to “usage of stoichiometric metal reductant; in high D-incorporation is still highly desirable.” “Initially, we started our investigation using D₂O as the economic deuterium source,” has been changed to past tense.

Thanks again for the professional and kind suggestions. We hope that this manuscript will be suitable for publication in *Nat. Commun.* after the revision.

Best regards,

Youai Qiu

REVIEWERS' COMMENTS

Reviewer #1 (Remarks to the Author):

The author addressed all the comments satisfactorily. The quality of the manuscript improved vastly than the earlier submission. Therefore, the manuscript can be accepted by Nat. Commun.

Reviewer #2 (Remarks to the Author):

After a second examination, this new version of the manuscript and the answers given by the authors are quite satisfying. I, therefore, recommend publication of this work in Nature Commun. after a minor modification of the introduction. I do still believe that it would be of high interest for the readership of Nature Comm. if the introduction mentioned the pioneering work of Murray, Renaud and Grimshaw in the 70's, which dealt with the deuteration of aryl halides, somewhat even more challenging partners.

J. R. Cockrell and R. W. Murray, J. Electrochem. Soc. 1972, 119, 849.

R. N. Renaud, Can. J. Chem. 1974, 52, 376–380.

J. Grimshaw and J. Trocha-Grimshaw, J. Chem. Soc. Perkin Trans. 2 1975, 215–218

Reviewer #3 (Remarks to the Author):

The authors have addressed the comments, and revised the manuscript and the supporting information. After detailed and careful reviewing, the quality of whole work was highly improved, and recommended to be published in Nature Communications.

Comments from the reviewer 2:

Question: I do still believe that it would be of high interest for the readership of Nature Comm. if the introduction mentioned the pioneering work of Murray, Renaud and Grimshaw in the 70's, which dealt with the deuteration of aryl halides, somewhat even more challenging partners.

J. R. Cockrell and R. W. Murray, *J. Electrochem. Soc.* 1972, 119, 849.

R. N. Renaud, *Can. J. Chem.* 1974, 52, 376–380.

J. Grimshaw and J. Trocha-Grimshaw, *J. Chem. Soc. Perkin Trans. 2* 1975, 215–218

Response: Thanks for the comments. The pioneering work of Murray, Renaud and Grimshaw has been cited as ref. [10], [11], and [12].

10. Cockrell, J. R., Murray, R. W. Deuterium labeling by electrochemical reactions. *J. Electrochem. Soc.*, **119**, 849–851 (1972).

11. Renaud, R. N. Electrochemical Synthesis of Deuterio Organic Compounds. I. Electrochemical Reduction of 1-Halonaphthalenes and Synthesis of 1-Chloro-4-methylnaphthalene. *Can. J. Chem.*, **52**, 376–380 (1974).

12. Grimshaw, J., Trocha-Grimshaw, J. Electrochemical reactions. Part XVIII. Reductive cleavage of aromatic carbon–halogen bonds in the presence of deuterium oxide. *J. Chem. Soc. Perkin Trans. 2*, 215–218 (1975).